# Targeting Angiogenesis in the Era of Biliary Tract Cancer Immunotherapy: Biological Rationale, Clinical Implications, and Future Research Avenues

**DOI:** 10.3390/cancers15082376

**Published:** 2023-04-19

**Authors:** Annalisa Schirizzi, Giampiero De Leonardis, Vincenza Lorusso, Rossella Donghia, Alessandro Rizzo, Simona Vallarelli, Carmela Ostuni, Laura Troiani, Ivan Roberto Lolli, Gianluigi Giannelli, Angela Dalia Ricci, Rosalba D’Alessandro, Claudio Lotesoriere

**Affiliations:** 1Laboratory of Experimental Oncology, National Institute of Gastroenterology—IRCCS “Saverio de Bellis”, 70013 Castellana Grotte, Italy; annalisa.schirizzi@irccsdebellis.it (A.S.); giampiero.deleonardis@irccsdebellis.it (G.D.L.); 2Clinical Trial Unit, National Institute of Gastroenterology—IRCCS “Saverio de Bellis”, 70013 Castellana Grotte, Italy; vincenza.lorusso@irccsdebellis.it; 3Data Science Unit, National Institute of Gastroenterology—IRCCS “Saverio de Bellis”, 70013 Castellana Grotte, Italy; rossella.donghia@irccsdebellis.it; 4Struttura Semplice Dipartimentale di Oncologia Medica per la Presa in Carico Globale del Paziente Oncologico “Don Tonino Bello”, I.R.C.C.S. Istituto Tumori “Giovanni Paolo II”, Viale Orazio Flacco 65, 70124 Bari, Italy; rizzo.alessandro179@gmail.com; 5Medical Oncology Unit, National Institute of Gastroenterology—IRCCS “Saverio de Bellis”, 70013 Castellana Grotte, Italy; simona.vallarelli@irccsdebellis.it (S.V.); carmela.ostuni@irccsdebellis.it (C.O.); laura.troiani@irccsdebellis.it (L.T.); ivan.lolli@irccsdebellis.it (I.R.L.); claudio.lotesoriere@irccsdebellis.it (C.L.); 6Scientific Direction, National Institute of Gastroenterology—IRCCS “Saverio de Bellis”, 70013 Castellana Grotte, Italy

**Keywords:** tumor microenvironment, immunotherapy, angiogenesis, VEGFA, combined therapy

## Abstract

**Simple Summary:**

The incidence of biliary tract cancers (BTCs) is rising. Although therapeutic weapons are still limited, a better knowledge of the molecular landscape of these tumors has established the conditions for several phase I and II clinical trials in which molecular targeted drugs are combined with chemotherapy. Clinical research is investing great efforts in immunotherapy, which has recently transformed the treatment of many cancers. In BTCs, the efficacy of immunotherapy is limited by the high percentage of patients presenting immunosuppressive features in the tumor microenvironment. Angiogenesis is known to play a key role in modulating the microenvironment by excluding cytotoxic immune cells from the tumor bed, favoring an immunosuppressive activity. Therefore, blocking angiogenesis could create the conditions for more efficacious immunotherapy. This review considers clinical trials based on therapy combining angiogenesis and immune cell inhibitors, highlighting the clinical need for phase III studies in a precision-medicine era.

**Abstract:**

Although biliary tract cancers are traditionally considered rare in Western countries, their incidence and mortality rates are rising worldwide. A better knowledge of the genomic landscape of these tumor types has broadened the number of molecular targeted therapies, including angiogenesis inhibitors. The role of immune checkpoint inhibitors (ICIs) could potentially change the first-line therapeutic approach, but monotherapy with ICIs has shown disappointing results in CCA. Several clinical trials are evaluating combination strategies that include immunotherapy together with other anticancer agents with a synergistic activity. The tumor microenvironment (TME) composition plays a pivotal role in the prognosis of BTC patients. The accumulation of immunosuppressive cell types, such as tumor-associated macrophages (TAMs) and regulatory T-cells, together with the poor infiltration of cytotoxic CD8+ T-cells, is known to predispose to a poor prognosis owing to the establishment of resistance mechanisms. Likewise, angiogenesis is recognized as a major player in modulating the TME in an immunosuppressive manner. This is the mechanistic rationale for combination treatment schemes blocking both immunity and angiogenesis. In this scenario, this review aims to provide an overview of the most recent completed or ongoing clinical trials combining immunotherapy and angiogenesis inhibitors with/without a chemotherapy backbone.

## 1. Introduction

Biliary tract cancer (BTC) is composed of a series of invasive gastrointestinal malignancies, mainly intrahepatic cholangiocarcinoma (iCCA), extrahepatic cholangiocarcinoma (eCCA), gallbladder carcinoma (GBC), and ampulla of Vater [1].

BTCs are classified based on the anatomic site inside the biliary tree; nevertheless, increasing evidence has demonstrated that they are a group of very heterogeneous tumors. Recent studies have highlighted molecular differences, in particular, between GBC and CCA and within CCAs [2]. BTCs are rare cancers, with an estimated incidence of <6 cases per 100,000 inhabitants, although the incidence is rising, mainly in parallel with the increased incidence of iCCA [3,4]. It is important to emphasize that surgery currently offers the only chance of cure for BTC patients and that the main goal of surgery is complete (R0) resection with low post-operative mortality [5], although, as with many other gastrointestinal tumors, most BTC patients are diagnosed at a later stage, so only 20% of BTC patients are eligible for surgical removal [6]. Even after radical surgery, the 1-year recurrence rate is about 50% [1,7]. Gemcitabine combined with cisplatin was the standard first-line treatment for advanced BTC until recently, according to the ABC-02 phase III clinical trial, but the improved overall survival (OS) was limited to less than 1 year [8]. Meanwhile, the latest findings from a TOPAZ-1 trial interim analysis reveal a potential paradigm shift in the first-line setting. This randomized phase III trial compared the combination of the PD-L1 inhibitor durvalumab plus gemcitabine—cisplatin (GemCis) versus standard-of-care doublet GemCis alone. The addition of the anti PD-L1 agent to the chemotherapy backbone determined a significant improvement in terms of median OS [9,10].

While immune checkpoint inhibitors (ICIs) have still to be found a niche in the BTC therapeutic algorithm, a wide spectrum of genomic alterations and possible druggable mutations has been investigated, including Isocitrate Dehydrogenase-1 (IDH-1) mutations and Fibroblast Growth Factor Receptor (FGFR) fusions and rearrangements, the BRAF proto-oncogene (BRAF), and HER-2 amplification neurotrophic tyrosine receptor kinase (NTRK) [11,12,13]. Targeted treatments, following a precision molecular characterization of BTC, especially iCCA, are entering into clinical practice thanks to recent results of studies exploring this therapeutic approach in BTC patients harboring specific genetic aberrations in second and later lines [7,12,14,15,16,17,18,19,20,21] (Figure 1).

The introduction of ICIs, such as anti-PD-1/PD-L1 and anti-CTL4, has profoundly transformed the therapeutic landscape of solid tumors. Despite their increasingly widespread employment, the mechanisms responsible for the onset of primary and secondary resistance to ICIs are poorly understood. Hence, specific and reliable predictors of tumor response to ICIs are still lacking [22]. In the attempt to unveil the determinants of tumor resistance, increasing interest has been focused on the tumor microenvironment (TME) [23,24,25]. Several studies suggest a crucial involvement of the TME—both paracrine and juxtacrine signaling—in tumorigenesis and in the development of ICI resistance [23,26,27,28]. Inflammation and inflammatory biomarkers are also important determinants in the tumor microenvironment, as numerous inflammatory cells and cytokines influence proliferation, angiogenesis, and metastasis [29,30]. Furthermore, either a hypoxic environment or the vascular endothelial growth factor (VEGF) secreted by tumor-associated cells induce PD-L1 upregulation and reduce the expansion of antitumor cells, such as CD8 + T-lymphocytes, natural killer (NK) cells, and M1-polarized TAMs, inducing the expansion of tumor-tolerant populations, such as M2-polarized TAMs, myeloid-derived suppressor cells (MDSC), and T-regulatory (Treg) cells, resulting in tumor immunoresistance [31,32,33]. Similarly, the intrinsic mechanisms underlying numerous oncogenic pathways, such as PI3K/Akt-, MEK/ERK-, and RAS-dependent signaling, are deemed to drive PD-L1 upregulation [34].

This review summarizes the current understanding of the role of angiogenesis as a predictive factor in immunotherapy resistance in BTC, focusing on the biological rationale and ongoing clinical trials.

## 2. Tumor Microenvironment in Biliary Tract Tumors

Compared to other tumors, the TME of BTC is not well characterized, although recent data indicate that most CCAs have either immune-excluded phenotypes (T-cell effectors confined to the tumor margin and unable to penetrate the tumor bed) or immune-deserted phenotypes (little or no T-cell infiltration in the stroma and tumor bed) [35]. Recent genomic analyses have defined that only a small percentage of BTC tumors are characterized by inflammation with high infiltration of effector and cytotoxic T cells and natural killer (NK) cells that mediate tumor immunosurveillance. Molecular characterization of this class of tumors has revealed a high incidence of mutational burden (TMB) often associated with mismatch repair (MMR) deficiency and high DNA microsatellite instability (MSI). These conditions, representing a prerequisite for the efficacy of ICI, were found in a minor percentage of the different BTC classes. Therefore, the composition of the TME is a key determinant of patient prognosis in BTC. The accumulation of immunosuppressive cell types, such as tumor-associated macrophages (TAMs) and regulatory T cells (T-reg), as well as poor infiltration of cytotoxic CD8+ T cells, and defective expression of MHC class 1 are all linked to poor prognosis and gemcitabine resistance [2,36].

Recent studies using single-cell RNA sequencing (scRNA-seq) of paired tumor and peritumor samples have provided a deeper insight into the features of immune cells infiltrating CCA, revealing an altered network of transcription factors in CCA-infiltrating T cells compared to peritumoral T cells. This has suggested impaired effector function by tumor-infiltrating CD8+ T cells and increased immunosuppression by CD4+ Tregs cells. Alvisi and colleagues used high-dimensionality flow cytometry to characterize T-cell and myeloid subsets in tumor tissues and their peritumoral and circulating tumor-free counterparts. Their results revealed that the TME of these tumors is characterized by a low infiltration of tumor-specific CD8+ T cells, marked by CD39+, which is associated with a high infiltration of hyper-activated CD4+ regulatory T cells (Tregs). Tregs frequency compared to that of CD4+ CD69+ T cells and conventional type 2 dendritic cells was linked to a poor prognosis [37]. In another study, Zhang et al. detected a diverse population of cancer-associated fibroblasts (CAFs), in which a predominant subpopulation of vascular CD146+ CAFs expressed high levels of IL-6, which in turn promoted tumor progression through IL-6R signaling on malignant cells [38]. It is widely accepted that the molecular heterogeneity of cells in iCCA tumors accounts for the remodeling of the TME and the decreased cytotoxicity of CD8+ T lymphocytes. In this scenario, Ma et al. hypothesize that hypoxia-induced VEGF expression results in the restriction of T-cell infiltration by transforming growth factor β (TGF-β), thus leading to a decreased responsiveness to immunotherapeutic treatments [39].

The VEGF/VEGFR axis is understood to play a primary role not only in the process of tumor growth and angiogenesis but also to be involved at multiple levels in the regulation of the cancer immune cycle, producing substantial changes that ultimately contribute to the creation of a microenvironment allowing the tumor to evade immune surveillance [40]. VEGFA alters the process of antigen presentation by impairing the maturation of DCs and inducing PD-L1 expression in tumor-associated myeloid DCs. VEGFA compromises proper hematopoiesis and makes thymocytes more susceptible to death by apoptosis, causing a suppression of the development and function of T lymphocytes. There is known to be a direct relationship between VEGFA secretion by tumor cells and increased amounts of Treg at the tumor site. The tumor vasculature may upregulate or downregulate proteins that control immune cell homing and trafficking, creating a selective immune cell barrier on the endothelium [41]. Accordingly, VEGF plays a key role in reprogramming the TME in an immunosuppressive manner, thus providing a mechanistic rationale for treatment schemes in which ICIs are combined with drugs that, by blocking VEGF signaling, increase the efficacy of the immunotherapy [42].

In order to decipher the TME heterogeneity of BTCs, increasingly, it is hoped that studies combining single-cell and spatial transcriptomics will be able to better define the intricate network of relationships between cells within the tumor and with the peritumor stroma, thus allowing the design of targeted therapeutic strategies [43].

## 3. Angiogenesis as a Hallmark of BTCs

BTCs are highly vascularized tumors characterized by an increased microvascular density (MVD).

Different studies have demonstrated that MVD is a hallmark of tumor neovascularization, and higher MVD is associated with an advanced tumor stage and low cure and high recurrence rates after surgical resection. Several signal cascades, including those regulated by VEGF, Platelet-derived growth factor (PDGF), and FGFR, play a crucial role in the pathogenesis and progression of BTCs [44]. Specifically, VEGF overexpression is observed in 42–76% of patients with unresectable advanced BTC and is associated with poor prognosis [14,45]. VEGFA overexpression is correlated with MVD and VEGF-C and -D, which are linked with lymph node metastases [46,47]. Boonjaraspinyo and colleagues reported that PDGF was overexpressed in cellular models of CCA and in human CCA samples (84.6%). Furthermore, the authors showed that PDGF expression is positively correlated with stage, metastasis, and short survival rate [48]. Results of another study showed that PDGF-D expression by CCA cells promotes the recruitment of cancer-associated fibroblasts (CAFs), thus mediating the tumor–microenvironment interplay [49]. The increasing progress of analyses aimed at characterizing the genomic landscape of BTC has revealed that several aberrations in multiple genes involved in angiogenesis, such as FGF receptor-2 (FGFR-2), are implicated in BTC [50]. Molecular characterization of a large cohort of Japanese BTC patients (260 cases, including iCCA, eCCA, and GBC) led to the identification of 32 significantly altered genes, including potentially druggable genetic alterations in 40% of cases. FGFR2 fusion genes were mainly found in iCCA cases, while gene fusions involving Protein Kinase CAMP-Activated Catalytic Subunit Alpha (PRKACA) or Protein Kinase CAMP-Activated Catalytic Subunit Beta (PRKACB) were preferentially found in eCCA [51]. Furthermore, genomic studies confirmed that molecular pathways that drive cell growth and angiogenesis, such as EGF, RAS, AKT, and MET, are often induced in BTCs [52]. The role of microRNAs (miRs), small non-coding RNAs, in promoting angiogenesis and thus tumorigenesis and the development of BTCs is proving to be no less important [53].

In summary, high vascularity, which characterizes all types of BTC and results from mutations in genes regulating angiogenesis, appears as a major driver of tumor aggressiveness and poor prognosis.

### Blocking Angiogenesis to Induce an Immunoresponsive Microenvironment: A Winning Strategy?

In BTC, several small trials of targeted therapy involving the VEGF pathway (Sorafenib, Cabozantinib, Bevacizumab, Sunitinib, or Axitinib) have been carried out but have showed only a modest efficacy [54,55,56,57]. Furthermore, FGFR plays an important role in the regulation of cell proliferation, migration, invasion, and angiogenesis in BTC. Previous studies have identified FGFR fusions in patients with BTC and have shown that the presence of these alterations predicts a more favorable prognosis [44]. Several clinical trials of drugs targeting FGFR fusions, including Infigratinib, Pemigatinib, and Derazantinib, have been conducted [58]. Given the modest and often controversial results obtained by therapies with single ICI or antiangiogenic agents in the treatment of BTC tumors, and considering the immunomodulatory effects of the latter, several ongoing/completed clinical trials have been based on the combination of immunotherapeutic and antiangiogenic agents (Figure 2).

Recent evidence obtained in mouse models of HCC has emphasized that the efficacy of combination treatment is due to a significant inhibition of immune suppression signaling, associated with TGFß, and the simultaneous induction of an immunoreactive microenvironment. The results of this study suggested that, based on immunocompetent genomic profiles in human HCC, 22% of patients were potential responders to combination therapies, with tumors characterized by Treg cell infiltrates, low inflammatory signaling, and activation of the VEGFR pathway [59].

## 4. Clinical Studies Targeting Both Angiogenesis and Immune Checkpoint Factors

Clinical trials have shown that treatment combining antiangiogenic drugs with an antibody directed against the immune checkpoint significantly increases survival compared to standard treatment in RCC, non-small cell lung cancer (NSCLC), and HCC. In BTCs there are few completed studies, and many are still in phase I or II. However, early findings suggest a clinical benefit and manageable safety profile of this therapeutic approach. Some of the clinical trials employ anti-VEGF antibodies, others Tyrosine Kinase inhibitors (TKs), and it is still unclear which class of antiangiogenic drugs works better in combination with immunotherapy. Antibodies are more specific, being directed against VEGF-A or VEGFR2, while the other agents affect a broad spectrum of tyrosine kinases, in addition to those regulating pro-angiogenic signaling. This overview of completed clinical studies, for which results have been published (Table 1), and those still ongoing (Table 2) shows that available findings indicate the efficacy and tolerability of combination therapies of immunotherapeutic and antiangiogenic agents, laying the groundwork for clinical trials with larger patient cohorts, possibly selected on the basis of specific markers.

We performed searches of Pubmed/Medline, the Cochrane library, and Scopus using the keywords “biliary tract cancer” OR “intrahepatic cholangiocarcinoma” OR “extrahepatic cholangiocarcinoma” OR “perihilar cholangiocarcinoma” OR “distal cholangiocarcinoma” OR “gallbladder carcinoma” OR “ampulla of Vater” AND “antiangiogenic therapy” AND “immunotherapy” OR “immune checkpoint inhibitors” OR “PD-L1 inhibitors” OR “immunotherapy and antiangiogenic combined therapy”. We selected the most relevant and pertinent studies, considering the quality of the studies in terms of their applicability, how they were conducted, the statistical analysis, the number of patients enrolled, and outcomes. For ongoing clinical trials, we searched the ClinicalTrials.gov database for recruiting and active, not recruiting trials, using the following keywords: “biliary tract cancer” OR “intrahepatic cholangiocarcinoma” OR “extrahepatic cholangiocarcinoma” OR “perihilar cholangiocarcinoma” OR “distal cholangiocarcinoma” OR “gallbladder carcinoma” OR “ampulla of Vater” AND “antiangiogenic therapy” AND “immunotherapy” OR “immune checkpoint inhibitors” OR “PD-L1 inhibitors” OR “immunotherapy and antiangiogenic combined therapy” OR “angiogenesis inhibitors and immunotherapy”. We restricted our research to phase I, II, or III clinical trials focused on the metastatic/advanced setting.

### 4.1. Targeting the VEGF Pathway

Bevacizumab is a VEGF monoclonal antibody approved for several solid tumors but not yet for treating BTCs. The triple combination therapy with atezolizumab (a PD-L1 inhibitor), bevacizumab, and platinum-based chemotherapy (carboplatin plus paclitaxel) was evaluated in phase III studies in patients with advanced NSCLC (IMpower 150) and ovarian carcinoma (IMagyn050), yielding positive results in the first case and negative results in the second, and good tolerability in both cases. These different results suggest that the response to this regimen may vary depending on the tumor, so its efficacy in BTC remains to be established [68,69]. IMbrave 151 is the first randomized trial to evaluate PD-L1/VEGF blockade combined with a chemotherapy backbone in BTC (NCT04677504). This double-blind, placebo-controlled, multicenter, international phase II study assesses the safety and efficacy of this combined scheme as first-line treatment for advanced BTC. In total, 162 patients were randomized to receive either atezolizumab + bevacizumab + CisGem (*n* = 79) or atezolizumab + placebo + CisGem (*n* = 83). The primary efficacy endpoint is progression-free survival (PFS). Partial results of this study were recently presented at the latest ASCO GI but showed only a modest increase in median PFS in the experimental arm of 8.4 months compared to 7.9 months in the control arm (HR, 0.76; 95% CI, 0.51–1.14). Despite these disappointing results and pending OS data, the study authors emphasize that the 6-month PFS rates were 78% and 63%, respectively [70]. The study also included exploratory analysis of specific biomarkers of response to therapy in tissue, blood, and stool samples collected at baseline and during treatment. Another ongoing study evaluating the efficacy of upfront integrated treatments is based on hepatic arterial infusion chemotherapy (HAIC), which delivers a high concentration of chemotherapeutic agents directly to liver tumors and has proven effective for intrahepatic and perihepatic cholangiocarcinoma. This phase II clinical trial (32 participants) aims to test the efficacy and safety of intra-arterial infusion of oxaliplatin, 5-fluorouracil, and bevacizumab combined with intravenous infusion of the PD-1 inhibitor (Toripalimab) in the treatment of unresectable biliary malignancies. This study has ORR and PFS as primary outcomes (NCT04217954). Other ongoing small studies based on the combination of bevacizumab with ICIs evaluate the efficacy and tolerability of this regimen in patients with BTC after first-line treatment. The phase Ib/II single-arm COMBATBIL study is testing the efficacy of mFOLFOX6, Bevacizumab, and Atezolizumab. The primary endpoint is the overall response rate (ORR) concurrently with disease control rate (DCR), PFS, overall survival (OS), and safety assessment (NCT05052099). Another ongoing phase II study (39 participants) is comparing bevacizumab, Durvalumab, Tremelimumab, and transarterial chemoembolization (TACE) versus Bevacizumab, Durvalumab, and Tremelimumab in patients with advanced HCC or BTC treated with (NCT03937830).

Ramucirumab is an IgG1 antagonist of VEGFR-2 that was tested in a cohort of 26 patients with advanced or metastatic BTC in a non-randomized phase I trial (JVDF) in combination with the anti-PD-1 antibody Pembrolizumab. This combination therapy was reported to have no serious adverse events and was associated with a PFS of 6.4 months compared to that of the control arm of 1.6 months [60].

### 4.2. Targeting TKs

Regorafenib is a small molecule that targets several protein kinases, including platelet-derived growth factor receptor (PDGFR), VEGFR fibroblast growth factor receptor (FGFR), FMS-like tyrosine kinase 3 (FLT-3), REarranged during Transfection (RET), and c-kit receptor tyrosine kinase (KIT). Regorafenib has shown clinical activity in patients with advanced BTC. Preclinical data suggested that this drug modulates antitumor immunity and exerts a synergistic action with immune checkpoint inhibition. REGOIMMUNE is a phase II, single-arm, multicenter basket study (NCT03475953). The regorafenib and Avelumab combination scheme was evaluated in several tumor types, including BTC; one cohort consisted of 34 patients with progression after first-line treatment. Best response was a partial response in four patients (13.8%), stable disease in 11 patients (37.9%), and progressive disease in 14 patients (48.3%). Median PFS and median OS were 2.5 months (95% CI) and 11.9 months (95% CI), respectively. Overexpression of PD-L1 and indoleamine 2,3-dioxygenase were associated with improved outcomes. Based on these results, further investigations in selected patients based on the characteristics of the TME are needed [61]. In another ongoing phase I/II clinical trial, the combination of Regorafenib and Durvalumab (MEDI4736) in patients with advanced BTC tumors is under investigation (NCT04781192).

Lenvatinib is a small-molecule tyrosine kinase inhibitor which inhibits several kinases implicated in BTC carcinogenesis, including VEGFR1-3, FGFR1-4, platelet-derived growth factor receptor a (PDGFRa), stem cell factor receptor (KIT), and rearranged transfection factor receptor (RET). Lenvatinib, due to its tolerability and versatility, has been widely used in combination therapy. Its efficacy has been demonstrated in tumor shrinkage in HCC [71], endometrial carcinoma [72,73], and renal cell carcinoma [74]. Lenvatinib has been proven to exert immunomodulatory effects. Indeed, combination treatment with lenvatinib plus pembrolizumab after first-line treatment progression was shown to be effective and tolerable for refractory BTC in a single-arm, investigator-initiated study. The results showed an ORR of 25%, a clinical benefit rate (CBR) of 40.5%, a PFS of 4.9 months, and a median OS of 11.0 months [62]. In a cohort of 38 patients with unresectable BTC enrolled in a phase II study, an ORR of 42.1% and a DCR of 76.3% were reported. Thirteen patients (34.2%) achieved downstaging and underwent surgery, six of whom (46.2%) achieved a major (*n* = 2) or partial (*n* = 4) pathological response in the primary tumor [63]. A real-world, retrospective Chinese study recruited 74 patients with BTC and tested the efficacy and safety of Lenvatinib TKI combined with PD-1 antibody after failure of first-line CisGem chemotherapy. The median number of PD-1 antibody cycles administered was 6.43 (95% CI: 5.83–7.04), and the median duration of Lenvatinib therapy was 21.0 weeks (95% CI: 18.04–23.93). Twenty-eight patients (37.83%) achieved an objective response according to the RECIST1.1, with a median follow-up duration of 15.0 months. The ORR was 20.27% (95% CI: 10.89–29.65%) and the DCR was 71.62% (95% CI: 61.11–82.14%). The median PFS and OS were 4.0 months (95% CI: 3.5–5.0) and 9.50 months (95% CI: 9.0–11.0), respectively. In an analysis performed in a subgroup of patients, tumor PD-L1 expression ≥50% and tumor mutation burden (TMB) ≥2.5 Muts/Mb were related to a longer PFS [64]. In a single-arm phase II clinical trial (NCT03951597) based on the combined treatment of a PD-1 antibody in combination with lenvatinib and GEMOX chemotherapy in the first-line treatment of unresectable advanced ICC, an ORR of 80% and a DCR of 93.3% (28/30) were observed. In addition, successful radical resection after downstaging was performed in three patients. These still-unpublished data, presented at ASCO 2021, suggest that the PD1 monoclonal antibody combined with lenvatinib and chemotherapy with GEMOx scheme may be an ideal conversion therapy for patients with potentially resectable advanced BTCs. Another multicenter phase II clinical trial (NCT05620498) is currently in progress assessing the efficacy of combining Tislelizumab, Lenvatinib, and the GEMOX scheme in potentially resectable advanced BTC conversion therapy. In a recently published case report, sequential perioperative therapy of four cycles of GemCis on days 1 and 8; lenvatinib from day 1 to day 21; and tislelizumab on day 15 was found to be effective and well-tolerated [75]. The LEAP-005 (NCT03797326) ongoing study is evaluating the efficacy and safety of Lenvatinib plus Pembrolizumab in patients with previously treated advanced solid tumors. Preliminary results obtained in 31 patients enrolled in the BTC cohort indicated an ORR of 10% (95% CI, 2–26) and a DCR of 68% (95% CI, 49–83). The median DOR was 5.3 months (range, 2.1 + to 6.2). The median PFS was 6.1 months (95% CI, 2.1–6.4). The median OS was 8.6 months (95% CI, 5.6 to NR). Treatment-related side effects occurred in 30 patients (97%), with grade 3–4 side effects in 15 (48%), but no treatment-related deaths were reported. Based on these encouraging data, enrollment in the BTC cohort has been further expanded. Another ongoing phase II study (NCT04211168) is assessing the efficacy and tolerability of combining Lenvatitinib with the anti-PD-1 Toripalimab as second-line treatment in patients with advanced BTC. The authors also aim to explore potential predictive biomarkers, but no results are available to date. A new phase I/II study (NCT05327582) was designed to evaluate the safety and efficacy (ORR) of the combined regimen (termed PLENA regimen) of Durvalumab (anti-PD-L1 antibody), Lenvatinib, and Paclitaxel albumin (nab-paclitaxel) in 65 inoperable patients with advanced BTC or pancreatic cancer.

Anlotinib is a novel oral tyrosine kinase inhibitor (TKI) targeting c-kit, PDGFR α/β, FGFR, and VEGFR and other important tyrosine kinase receptors. In a phase Ib study, sintilimab (a fully human IgG4 anti-PD-1 monoclonal antibody) plus first-line anlotinib demonstrated efficacy, durability, and tolerability in patients with NSCLC, showing that this may be a new chemo-free regimen [76]. Nevertheless, the feasibility and safety of the antiangiogenic tyrosinase inhibitor plus PD-1 blockade for patients with previously treated advanced BTC are still limited. Encouraging results have been reported for a phase II study evaluating the efficacy and safety of sintilimab plus anlotinib as second-line treatment for patients with advanced BTC. This combination regimen of sintilimab plus anlotinib showed an mOS of 12.3 months with a 12-month OS rate of 55.0%, an mPFS of 6.5 months, an ORR of 30%, and a DCR of 95%. The authors also investigated potential prognostic biomarkers, including PD-L1 status and the somatic mutation profile, as well as the gut microbiota, which is believed to be associated with the response to immunotherapy. The analysis presented in this study showed that an enrichment of the gut microbiome in proteobacteria was associated with a lower clinical response [66]. Another phase Ib study evaluated the efficacy and safety of anlotinib plus TQB2450, a programmed death-ligand 1 inhibitor in pre-treated advanced BTCs. The results indicated an objective response rate of 21.21%, a disease control rate (DCR) of 72.73%, and a clinical benefit rate (CBR) of 42.42%. With a median follow-up of 19.68 months, median progression-free survival (PFS) and overall survival (OS) were 6.24 (95% confidence interval (CI), 4.11–8.25) and 15.77 (95% CI, 10.74–19.71) months, respectively. In addition, the authors indicated the KRAS mutation and high TMB as possible biomarkers of treatment efficacy [65].

Axitinib is a potent and selective tyrosine kinase inhibitor that targets vascular endothelial growth factor receptors (VEGFR)-1, VEGFR-2, and VEGFR-3. One phase II study is planned to enroll 60 patients within hepatobiliary malignant tumors, including BTC subjects (NCT04010071). Primary outcomes include ORR and PFS in second-line therapy combining axitinib plus toripalimab (anti PD-1).

Cobimetinib is an MEK inhibitor known to modulate the TME of preclinical models of colon cancer, breast cancer, and melanoma [77,78,79], thus exerting a combined synergistic action with PD-1 and PDL-1 inhibitors. Although good tolerability and promising clinical activity, combining an MEK inhibitor with PD-L1 inhibition, were observed in phase I studies in colorectal cancer, these results were not confirmed in a confirmatory phase III clinical trial in colorectal cancer [80]. Nevertheless, a recent phase II study of 77 patients with advanced BTC already treated with first-line or subsequent therapies, evaluating the efficacy of a combination therapy of atezolizumab (anti-PDL-1) with or without cobimetinib, reported good results in terms of safety and a median PFS of 3.65 months in the combination arm versus 1.87 months in the monotherapy arm (HR 0.58, 90% CI 0.35–0.93, one-tail *p* = 0.027). One patient in the combination arm (3.3%) and one patient in the monotherapy arm (2.8%) had a partial response. Although this study indicates that MEK inhibitors may have some benefit in the context of systemic immunotherapy for BTC, the modest increase in PFS observed probably reflects the additive effects of the individual therapies rather than a synergistic action resulting from MEK immunomodulation of the tumor microenvironment [67]. Finally, a phase II study is investigating the effect of combining two iICIs, Atezolizumab and CDX-1127 (varlilumab), with or without cobimetinib in BTC patients. Primary endpoints of the study are ORR and PFS in an estimated cohort of 64 patients (NCT04941287).

## 5. Conclusions and Future Directions

The results of the recently published TOPAZ-1 trial have paved the way for a new era in BTC. This double blind, placebo-controlled phase III study showed that the addition of durvalumab to gemcitabine/cisplatin resulted in a significantly improved OS. However, despite recent advances in systemic treatments and multimodal therapies, the prognosis for BTC remains poor. As molecular characterization and anticancer treatments proceed, the “one size fits all” approach may no longer be applied in BTC for several reasons, including the marked genetic heterogeneity of different subtypes, as well as recurrence patterns and unique biological and molecular profiles. In any case, immunotherapy seems to have finally found its “therapeutic niche” in BTC, and several ongoing studies are further investigating the potential role of combinatorial treatments with ICIs plus other anticancer treatments [67,81]. Among these, antiangiogenic therapies have been suggested to modify the TME by increasing the infiltration of T cells and so “boosting” an anticancer immune response. Given this evolving scenario, the scientific community is being unanimously driven to continue molecular testing at disease diagnosis. Continued progress in the field of BTC will demand further international, multicenter clinical trials, including genomic analysis of serial tumors, as well as the use of circulating tumor DNA samples during anticancer treatment. These approaches are compelling and may hopefully succeed in elucidating why ICIs and immune-based combinations may be ineffective in some BTC patients as well as suggesting new ways to overcome resistance to anticancer treatments in this aggressive, heterogeneous group of malignancies.

## Figures and Tables

**Figure 1 cancers-15-02376-f001:**
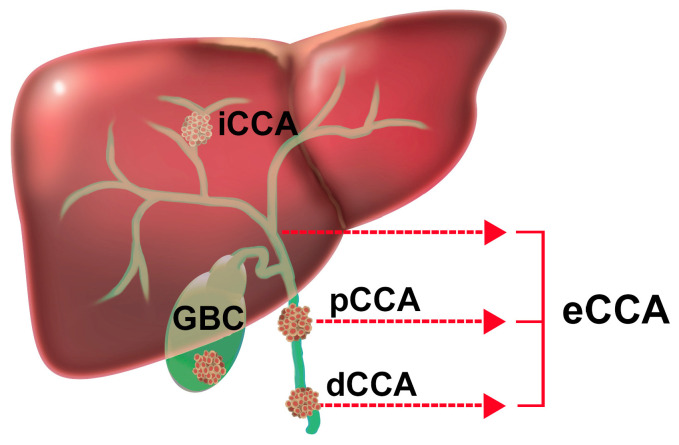
Schematic figure representing biliary tract cancer anatomical subgroups. Abbreviations: eCCA, extrahepatic cholangiocarcinoma; dCCA, distal cholangiocarcinoma; pCCA, perihilar cholangiocarcinoma; iCCA, intrahepatic cholangiocarcinoma; GBC, gallbladder cancer; ICI, immunocheckpoint inhibitors.

**Figure 2 cancers-15-02376-f002:**
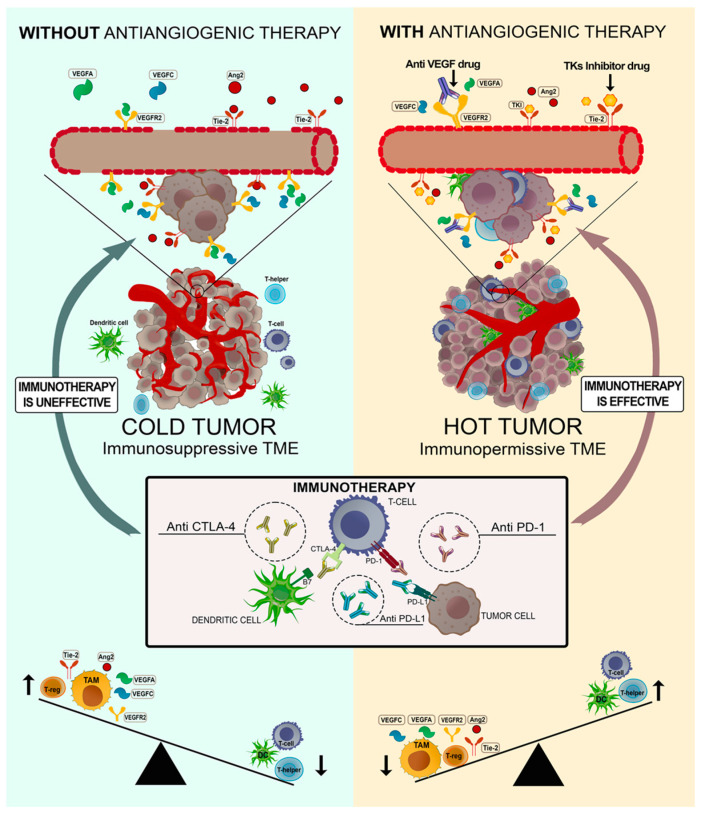
Schematic representation of the effects of antiangiogenic therapies on the conversion of an immunosuppressive (in blue) into an immunopermissive tumor microenvironment (in yellow). The immunosuppressive environment is characterized by fenestrated (**above**) and highly disorganized vessels (**middle**) and high levels of TAM, T-reg, Tie-2, Ang2, VEGFA, VEGFC, and VEGFR2 and low levels of T, T-helper, and dendritic cells (**bottom**), while the immunopermissive environment is characterized by the normalization of vessels and the inversion of expression levels of the same factors. This switch makes immunotherapy effective in highly vascularized BTCs. Abbreviations: TAM, tumor-associated macrophages; BTCs, biliary tract cancers; TME, tumor microenvironment.

**Table 1 cancers-15-02376-t001:** Comprehensive table of studies assessing immunotherapy and antiangiogenic combined therapy with reported results in biliary tract cancer. Abbreviations: BTC, biliary tract cancer; aBTC, advanced biliary tract cancer; CCA, cholangiocarcinoma; GBC, gallbladder cancer; PFS, progression-free survival; OS, overall survival; CBR, clinical benefit rate; DCR, disease control rate; ORR, overall response rate.

Phase	Setting	Treatment	Number of Patients	Type of Tumors	Results
I [60]	Second line	Ramucirumab + Pembrolizumab vs. Pembrolizumab	26 patients	aBTC	No adverse events; PFS of 6.4 months vs. 1.6 months
II [61]	Second line	Regorafenib + Avelumab	34 patients	BTC	13.8% partial response, 37.9% stable disease, and 48.3% progressive disease. Median PFS: 2.5 months; median OS: 11.9 months
II [62]	Second line	Lenvatinib + Pembrolizumab	50 patients	Advanced hepatobiliary malignancies	ORR of 25%, CBR of 40.5%, mPFS of 4.9 months, and mOS of 11.0 months
II [63]	First line	Lenvatinib +PD-1 inhibitors	38 patients	Unresectable BTC	ORR of 42.1% and DCR of 76.3%
Retrospective study [64]	Second line	Lenvatinib + Programmed Cell Death Protein-1 inhibitor	74 patients	BTC	ORR of 20.27%, DCR of 71.62%, mPFS of 4.0 months, and OS of 9.50 months
Ib [65]	Second line	TQB2450 + Anlotinib	66 patients	Advanced CCA	ORR of 21.21%, DCR of 72.73%, CBR of 42.42%, mPFS of 6.24 months, and OS of 15.77 months
II [66]	Second line	Anlotinib + Sintlimab	20 patients	aBTC	mOS of 12.3 months, 12-month OS rate of 55.0%, mPFS of 6.5 months, ORR of 30%, and DCR of 95%
II [67]	Second or later line	Atezolizumab + Cobimetinib vs. Atezolizumab	77 patients	BTC	mPFS of 3.65 months vs. 1.87 months

**Table 2 cancers-15-02376-t002:** Comprehensive table of ongoing trials evaluating immunotherapy in combination with antiangiogenic therapy in biliary tract cancer. Ongoing clinical trials were identified and selected by searching ClinicalTrials.gov. Abbreviations: aBTC, advanced biliary tract cancer; HCC, hepatocellular carcinoma; CRC, colorectal cancer; GIST, gastrointestinal stromal tumor; GOJ, gastroesophageal junction carcinoma; GC, gastric cancer; STS, soft-tissue sarcoma; RR-DTC, radioiodine-refractory differentiated thyroid cancer; GEP-NETs, neuroendocrine gastroenteropancreatic tumors; NSCLC, non-small cell lung cancer; ICC, intrahepatic cholangiocarcinoma; GBC, gallbladder cancer; TNBC, triple-negative breast cancer; OVC, ovarian cancer; GBM, glioblastoma biliary tract cancer; PAC, pancreatic cancer; PFS, progression-free survival; ORR, overall response rate; RP2D, recommended phase 2 dose; AEs, adverse events.

NCT (ClinicalTrial.gov)	Phase	Setting	Treatment	Number of Patients	Types of Tumors	Primary Outcomes	Status	Estimated Study Completion Date
NCT04677504 (IMbrave 151)	II	First line	Atezolizumab and GemCis + Bevacizumab vs. Atezolizumab, GemCis	162 patients	aBTC	PFS	Active, not recruiting	24 April 2023
NCT04217954	II	First line	HAIC with Oxaliplatin, 5-FU, and Bevacizumab plus intravenous Toripalimab	32 patients	aBTC	PFS, ORR	Recruiting	28 July 2023
NCT05052099	I/II	Second line	mFOLFOX6, Bevacizumab, and Atezolizumab	35 patients	aBTC	ORR	Recruiting	June 2024
NCT03937830	II	Second or later line	Arm A: Durvalumab + bevacizumab + Tremelimumab; Arm B: Durvalumab + bevacizumab + Tremelimumab + TACE	39 patients	HCC, BTC	PFS	Recruiting	31 December 2023
NCT03475953	I/II	Second or later line	Regorafenib + Avelumab	482 patients	Not MSI-H or MMR-deficient CRC, GIST, GOJ or GC, BTC, HCC, STS, RR-DTC, GEP-NETs, NSCLC, solid tumor with immune signature (TLS+)	ORR	Recruiting	31 December 2022
NCT04781192	I/II	Second or later line	Durvalumab + Regorafenib	40 patients	aBTC	AEs, PFS	Recruiting	December 2023
NCT05620498	II	Conversion therapy	Tislelizumab + Lenvatinib and GEMOX vs. Tislelizumab plus GEMOX	60 patients	ICC, GBC	ORR	Recruiting	31 March 2024
NCT03797326	II	Second line	Pembrolizumab + lenvatinib	590 patients	TNBC, OVC, GC, CRC, GBM, PAC	ORR, AEs	Active, not recruiting	22 December 2023
NCT04211168	II	Second line	Toripalimab + Lenvatinib	44 patients	aBTC	ORR, AE	Unknown	December 2022
NCT05327582	I/II	First or later line	Durvalumab + Lenvatinib plus Nab paclitaxel	65 patients	PC, BTC	AEs, PFS	Recruiting	30 April 2024
NCT04010071	II	Second line	Axitinib + Toripalimab	60 patients	Hepatobiliary neoplasm, liver neoplasm, BTC	ORR, PFS	Recruiting	18 August 2021
NCT04941287	II	Second line	Atezolizumab and CDX-1127 (Varlilumab) + Cobimetinib vs. Atezolizumab and CDX-1127	64 patients	aBTC	ORR, PFS	Recruiting	1 September 2023

## Data Availability

No new data were created.

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
