# Peer review of "Targeting Angiogenesis in the Era of Biliary Tract Cancer Immunotherapy: Biological Rationale, Clinical Implications, and Future Research Avenues"

_cancers, 2023, doi:10.3390/cancers15082376_

Round 1

Reviewer 1 Report

This is a comprehensive review, but there are still some shortcomings.

First, in section 2 "Tumor Microenvironment in Biliary Tract Tumors," there is a lack of examples to support your claims. For instance, on line 115, you describe the important impact of immunosuppressive cell types on BTC, and you also list some cell types, but there is a lack of substantive literature evidence to explain them further. Additionally, on line 120, there is no clear causal relationship stated, and there may be a description error.

Second, the second and third sections' conclusion lacks clear summaries of the key points discussed. Please emphasize the main points of each paragraph.

Third, please summarize the research methods and major findings used in the basic research discussed. There is an issue with incomplete descriptions of the methods used.

Finally, there are several mentions of the tumor microenvironment and tumor heterogeneity. It is well-known that single-cell and spatial omics are important tools for studying these aspects, but you have hardly mentioned them. Please consider adding appropriate information to the text and conclusion to address this issue.

Author Response

We have taken the reviewers' comments into account throughout the manuscript. All changes we have made have been indicated in blue font and we believe they have greatly improved the manuscript quality. In addition, as requested by one of the reviewers, the manuscript was proofread by a native English speaker and corrections were highlighted in red.

We hope that the manuscript in its current form can be accepted for publication.

Reviewer 1

This is a comprehensive review, but there are still some shortcomings.

We would like to thank the reviewer for his timely review and his suggestions which aimed to improve the quality of our study and which we have therefore accepted in full.

First, in section 2 "Tumor Microenvironment in Biliary Tract Tumors," there is a lack of examples to support your claims. For instance, on line 115, you describe the important impact of immunosuppressive cell types on BTC, and you also list some cell types, but there is a lack of substantive literature evidence to explain them further. Additionally, on line 120, there is no clear causal relationship stated, and there may be a description error.

As suggested by the reviewer, in section 2 we extensively argued the importance of the TME immunophenotype in BTCs and introduced the biological rationale for a therapy combining ICIs with anti-angiogenic drugs.

Second, the second and third sections' conclusion lacks clear summaries of the key points discussed. Please emphasize the main points of each paragraph.

Following the reviewer's suggestion, we underlined the main points discussed in paragraphs 2 and 3.

Third, please summarize the research methods and major findings used in the basic research discussed. There is an issue with incomplete descriptions of the methods used.

As the reviewer rightly pointed out, we have described in detail the methodology used in the research of the clinical studies presented in section 4.

Finally, there are several mentions of the tumor microenvironment and tumor heterogeneity. It is well-known that single-cell and spatial omics are important tools for studying these aspects, but you have hardly mentioned them. Please consider adding appropriate information to the text and conclusion to address this issue.

In the revision of section 2, we introduced the molecular studies that, thanks to new single-cell sequencing technologies, are contributing to the understanding of the immunophenotypic pattern of the TME of BTCs and how specific cell populations are able to reprogramme the TME and drive the response to a given drug therapy.

Reviewer 2 Report

I have read with some interest this manuscript, which is a review manuscript on the current knowledge, both basic-translational and clinic, on angiogenesis as a target to treat BTC. The manuscript is overall well written and sufficiently readable. I would suggest the following changes:

1. Please define much better the ideal patient to be considered for systemic target therapy. Considering that the current guidelines recommend surgery for “resectable patients”, I would ask to add one or two sentences to clarify that surgery is still the main therapy for resectable patients. What is resectable is another issue that matters HPB surgeons and MDT tumor boards. Please consider adding the following reference: 

Cillo U, et al. Surgery for cholangiocarcinoma. Liver Int. 2019 May;39 Suppl 1(Suppl Suppl1):143-155. doi: 10.1111/liv.14089. PMID: 30843343; PMCID: PMC6563077.

2. Another point that should be better stated is that the category “BTC” includes at least 4 different types of tumors (intrahepatic CCA; perihilar CCA; GBCA; dCCA). They are a very heterogenous group of tumors, which in clinical trials are barely separated. To me, this continuous to be a kind of bias. Please consider adding the following reference:

Malenica I. Molecular and Immunological Characterization of Biliary Tract Cancers: A Paradigm Shift Towards a Personalized Medicine. Cancers (Basel). 2020 Aug 6;12(8):2190. doi: 10.3390/cancers12082190. PMID:32781527; PMCID: PMC7464597.

3. Speaking about immunosuppressive TME in BTC, you should add this paper:

Alvisi G. Multimodal single-cell profiling of intrahepatic cholangiocarcinoma defines hyperactivated Tregs as a potential therapeutic target. J Hepatol. 2022 Nov;77(5):1359-1372. doi: 10.1016/j.jhep.2022.05.043.

Author Response

We have taken the reviewers' comments into account throughout the manuscript. All changes we have made have been indicated in blue font and we believe they have greatly improved the manuscript quality. In addition, as requested by one of the reviewers, the manuscript was proofread by a native English speaker and corrections were highlighted in red.

We hope that the manuscript in its current form can be accepted for publication.

Reviewer 2

I have read with some interest this manuscript, which is a review manuscript on the current knowledge, both basic-translational and clinic, on angiogenesis as a target to treat BTC. The manuscript is overall well written and sufficiently readable. I would suggest the following changes:

We thank the reviewer for time spent reviewing the manuscript and fully accept his suggestions. 

  1. Please define much better the ideal patient to be considered for systemic target therapy. Considering that the current guidelines recommend surgery for “resectable patients”, I would ask to add one or two sentences to clarify that surgery is still the main therapy for resectable patients. What is resectable is another issue that matters HPB surgeons and MDT tumor boards. Please consider adding the following reference:

Cillo U, et al. Surgery for cholangiocarcinoma. Liver Int. 2019 May;39 Suppl 1(Suppl Suppl1):143-155. doi: 10.1111/liv.14089. PMID: 30843343; PMCID: PMC6563077.

Following the reviewer's suggestion, we have included in the introduction paragraph the reference indicated, which emphasises the major role of surgery in the treatment of BTCs.

  1. Another point that should be better stated is that the category “BTC” includes at least 4 different types of tumors (intrahepatic CCA; perihilar CCA; GBCA; dCCA). They are a very heterogenous group of tumors, which in clinical trials are barely separated. To me, this continuous to be a kind of bias. Please consider adding the following reference:

Malenica I. Molecular and Immunological Characterization of Biliary Tract Cancers: A Paradigm Shift Towards a Personalized Medicine. Cancers (Basel). 2020 Aug 6;12(8):2190. doi: 10.3390/cancers12082190. PMID:32781527; PMCID: PMC7464597.

As suggested, we added the indicated reference in the introduction paragraph, to emphasise the heterogeneity of the different tumour types belonging to the class of BTCs

  1. Speaking about immunosuppressive TME in BTC, you should add this paper:

Alvisi G. Multimodal single-cell profiling of intrahepatic cholangiocarcinoma defines hyperactivated Tregs as a potential therapeutic target. J Hepatol. 2022 Nov;77(5):1359-1372. doi: 10.1016/j.jhep.2022.05.043.

As suggested by the reviewer, in section 2 we extensively argued the importance of the TME immunophenotype in BTCs and we added the indicated reference.

Reviewer 3 Report

Excellent review with insight into angiogenesis VEGF and immune modulation of biliary tract cancers.  Up-to-date review

Author Response

We have taken the reviewers' comments into account throughout the manuscript. All changes we have made have been indicated in blue font and we believe they have greatly improved the manuscript quality. In addition, as requested by one of the reviewers, the manuscript was proofread by a native English speaker and corrections were highlighted in red.

We hope that the manuscript in its current form can be accepted for publication.

Reviewer 3

Excellent review with insight into angiogenesis VEGF and immune modulation of biliary tract cancers.  Up-to-date review

We are grateful to the reviewer for his overwhelmingly positive assessment and hope that he will rate the changes made according to the reviewers' recommendations as positive.
